# BMS-986158, a Small Molecule Inhibitor of the Bromodomain and Extraterminal Domain Proteins, in Patients with Selected Advanced Solid Tumors: Results from a Phase 1/2a Trial

**DOI:** 10.3390/cancers14174079

**Published:** 2022-08-23

**Authors:** John Hilton, Mihaela Cristea, Sophie Postel-Vinay, Capucine Baldini, Mark Voskoboynik, William Edenfield, Geoffrey I. Shapiro, Michael L. Cheng, Jacqueline Vuky, Bradley Corr, Sharmila Das, Abraham Apfel, Ke Xu, Martin Kozicki, Keziban Ünsal-Kaçmaz, Amy Hammell, Guan Wang, Palanikumar Ravindran, Georgia Kollia, Oriana Esposito, Shodeinde Coker, Jennifer R. Diamond

**Affiliations:** 1Division of Medical Oncology, Ottawa Hospital, Ottawa, ON K1H 8L6, Canada; 2Department of Medical Oncology & Therapeutics Research, City of Hope National Medical Center, Duarte, CA 91010, USA; 3Drug Development Department, Institut Gustave Roussy, 94805 Villejuif, France; 4Department of Medical Oncology, Alfred Health, Melbourne 3004, Australia; 5Central Clinical School, Monash University, Melbourne 3800, Australia; 6Prisma Health Cancer Institute, Greenville, SC 29605, USA; 7Dana-Farber Cancer Institute, Boston, MA 02215, USA; 8Department of Medicine/Oncology, Oregon Health & Science University, Portland, OR 97239, USA; 9Division of Gynecologic Oncology, Department of Obstetrics and Gynecology, University of Colorado Anschutz Medical Campus, Aurora, CO 80045, USA; 10Bristol Myers Squibb, Princeton, NJ 08648, USA

**Keywords:** bromodomain, BET inhibitor, dose escalation, BRD2, BRD3, BRD4, NUT carcinoma, pan tumor

## Abstract

**Simple Summary:**

Bromodomain and extraterminal domain (BET) proteins can regulate cancer-related genes to make tumors grow. BMS-986158, an experimental anticancer therapy, blocks BET protein function. We researched BMS-986158′s potential anticancer effects, side effects, what happens to BMS-986158 in the body, and whether treatment causes any changes in gene activity. To determine the proper dose, 5 different doses were tested using 3 schedules (days taking and not taking therapy). Among 83 patients across all dose schedules, BMS-986158 was generally well tolerated. The most common side effects were diarrhea (43% of patients) and lower platelet cell numbers (blood clotting cells; 39%). Schedule A (cycles of 5 days on and 2 days off therapy) provided stable levels of BMS-986158 in the blood and showed some preliminary anticancer effects, with 30% of patients showing a clinical benefit (a period without meaningful tumor growth in 12 patients and tumor shrinkage by at least 30% in 2 patients).

**Abstract:**

This phase 1/2a, open-label study (NCT02419417) evaluated the safety, tolerability, pharmacokinetics (PK), and pharmacodynamics of BMS-986158, a selective bromodomain and extraterminal domain (BET) inhibitor. Dose escalation was performed with 3 BMS-986158 dosing schedules: A (5 days on, 2 days off; range, 0.75–4.5 mg), B (14 days on, 7 days off; 2.0–3.0 mg), and C (7 days on, 14 days off; 2.0–4.5 mg). Eighty-three patients were enrolled and received ≥1 BMS-986158 dose. Diarrhea (43%) and thrombocytopenia (39%) were the most common treatment-related adverse events (TRAEs). A lower incidence of TRAEs was found with schedules A (72%) and C (72%) vs. B (100%). Stable disease was achieved in 12 (26.1%), 3 (37.5%), and 9 (31.0%) patients on schedules A, B, and C, respectively. Two patients on schedule A with a 4.5-mg starting dose (ovarian cancer, n = 1; nuclear protein in testis [NUT] carcinoma, n = 1) experienced a partial response. BMS-986158 demonstrated rapid-to-moderate absorption (median time to maximum observed plasma concentration, 1–4 h). As expected with an epigenetic modifier, expression changes in select BET-regulated genes occurred with BMS-986158 treatment. Schedule A dosing (5 days on, 2 days off) yielded tolerable safety, preliminary antitumor activity, and a dose-proportional PK profile.

## 1. Introduction

Transcriptional dysregulation and genome instability caused by mutations and epigenetic alterations are common in cancer cells [1,2]. Epigenetic modulators present a novel strategy to combat transcriptional dysregulation, particularly at hyperactivated oncogenes such as *MYC* [1,2,3,4]. Bromodomain and extraterminal domain (BET) proteins (including bromodomain-containing [BRD] proteins BRD2, BRD3, BRD4, and BRD testis-specific [BRDT] protein) act as readers of acetylated lysine histone residues and recruit regulatory complexes for transcriptional modulation [3,5,6]. BRD4 preferentially binds to super enhancers of growth-stimulating promoters, such as at the oncogene *MYC*, which may allow cancer cells to be specifically sensitive to BET inhibitor therapy [3,4,7].

In addition to controlling expression of key oncogenes, *BRD4* mutations have also been documented in several cancers. Gene fusions of nuclear protein in testis (*NUT*) and *BRD4*, *BRD3*, or other genes are a characteristic of NUT carcinoma [8,9]. *BRD4* transcriptional dysregulation has been correlated with disease progression in several solid tumors [10,11,12]. In preclinical studies, cells with dysregulated *BRD* or *MYC* expression [13], specific mutations (e.g., *SWI/SNF*, *KRAS*, *Gnaq/II*) [14,15,16], or translocations (e.g., *BRD3-*, *4-NUT* or *EWS-FLI1* fusions) [17,18] were sensitive to BET inhibitors.

The BET inhibitor, BMS-986158, has demonstrated potent inhibition of BRD4 and *C-MYC* expression with favorable interactions in a water environment due to its novel carboline structure; it was identified and selected for investigation due to its potent preclinical cytotoxic effects against hematologic malignancies and solid tumors [19]. In mouse studies, low doses of BMS-986158 demonstrated antitumor activity against 9 of 19 (47%) of the patient-derived xenograft models (including ovarian cancer, lung squamous cell carcinoma, lung adenocarcinoma, colorectal cancer, and triple-negative breast cancer [TNBC]) [19].

In this study, we evaluated the safety, pharmacokinetics (PK), pharmacodynamics, and preliminary efficacy of BMS-986158 in patients with advanced solid tumors.

## 2. Materials and Methods

### 2.1. Study Design

CA011-001 was an open-label study of BMS-986158 monotherapy. Part 1 of the study included a dose-escalation segment and Part 2 was a dose-expansion phase. In Part 1, escalation occurred with an initial dose of BMS-986158 and increased in prespecified increments above the previous dose level until the first occurrence of any protocol-defined dose-limiting toxicity (DLT). Overall, 3 dosing schedules were evaluated to test safety and efficacy (Appendix A). Each patient on schedules A, B, and C was administered a single oral dose of BMS-986158 on cycle 1 day 1 (C1D1), and no additional doses were administered until cycle 2 day 1 (C2D1; ≈7 days after C1D1). Beginning on C2D1 and for each subsequent cycle, patients received treatment according to the following schedules. On schedule A, BMS-986158 was administered once daily (QD) for 5 days on and 2 days off with 0.75-, 1.25-, 2.0-, 3.0-, and 4.5-mg escalating doses. On schedule B, patients received BMS-986158 QD for 14 days on and 7 days off with 2.0- and 3.0-mg doses. On schedule C, BMS-986158 QD was given for 7 days on and 14 days off with 2.0-, 3.0-, and 4.5-mg doses. Dose escalation was guided by a modified toxicity probability interval (mTPI) design [20]. In the mTPI design, 3 to 4 patients would be enrolled at each dose level; if a DLT was observed, an additional 3 to 4 patients would be evaluated at the same dose level, up to a total of 13 patients. If no DLTs occurred, 3 to 4 patients would be enrolled at the next dose level. The target DLT rate was 27% to determine the maximum tolerated dose (MTD). For Part 2 of this trial (the dose-expansion phase), schedule A dosing was selected based on safety, pharmacodynamics, and PK data collected in Part 1. The dosing regimen for Part 2 was 4.5 mg for 2 weeks (10 doses) followed by 3.75 mg for the remainder of treatment. However, because study CA011-001 was terminated early with 1 patient enrolled in Part 2, we have grouped this patient with the schedule A, 4.5-mg cohort in Part 1 for the remainder of this report.

This study was performed in accordance with Good Clinical Practice as defined by the International Conference on Harmonization. Patients provided written informed consent, and the study was conducted with oversight from the Institutional Review Board/Independent Ethics Committee at each study site.

### 2.2. Patients

Eligible patients had ≥1 measurable lesion at baseline per Response Evaluation Criteria in Solid Tumors (RECIST) version 1.1; archival tumor tissues or consent to pretreatment biopsy; a life expectancy of ≥3 months; and Eastern Cooperative Oncology Group performance status 0–1; and had received standard-of-care treatments. During dose escalation, eligible patients included those who had ovarian cancer, SCLC, TNBC, or other selected solid tumors and had received at least 1 prior line of chemotherapy. Patients with ovarian cancer or SCLC must have received at least 1 prior platinum treatment (PT). Adolescents ≥12 years of age, who had NUT carcinoma or Ewing sarcoma, were eligible for enrollment. Patients with all other cancers were required to be ≥18 years of age. Patients were excluded from enrollment if they had concomitant malignancies (except adequately treated non-melanoma skin cancers or in situ bladder, breast, or cervical cancers), uncontrolled or significant cardiovascular disease, inadequate bone marrow function, chronic gastrointestinal illnesses, and/or prior BET inhibitor treatments. For Part 2, inclusion criteria allowed patients harboring genetic alterations in fusion proteins (NUT carcinoma, double-hit lymphoma) or gene amplifications *in BRD* (TNBC), *MYC* (non-germinal center diffuse large B-cell lymphoma), or *AR* (castration-resistant prostate cancer).

### 2.3. Study Endpoints and Assessments

The primary endpoints of the study were safety and tolerability as well as identification of DLTs, the MTD, and the recommended phase 2 dose (RP2D). Secondary and exploratory endpoints included the PK of BMS-986158 and its major oxidized metabolite, QTc dose–response and exposure-response effects measured on electrocardiograms (ECGs), and preliminary antitumor activity (best overall response, objective response rate [ORR], progression-free survival [PFS], and overall survival [OS]). Key exploratory endpoints included the associations of gene amplifications, mutations, and translocations with antitumor efficacy and pharmacodynamic effects on gene expression for BET-regulated genes.

Safety was assessed as adverse events (AEs) coded by the Medical Dictionary for Regulatory Activities version 24.0 and the National Cancer Institute’s Common Terminology Criteria for Adverse Events version 4.03. In addition, AEs were classified by whether they were considered possibly or probably treatment related (TRAEs) by investigator assessment, qualified as a serious AE, or led to interruption or discontinuation of study therapy.

At C1D1 in every dosing schedule cohort, a single dose was administered followed by 168 h (i.e., 7 days) of intensive PK sampling. Following cycle 1, the multiple-dose treatment schedules (described above) began in cycle 2. PK parameters were derived following single dose combined across all schedules and multiple doses at cycle 2 for each schedule. PK parameters included maximum observed plasma concentration (C_max_), time to C_max_ (T_max_), area under the plasma concentration-time curve from time 0 to time of the last quantifiable concentration (AUC_0–T_) or time 0 to time 24 h post dose (AUC_0–24_), half-life (T_1/2_), apparent total body clearance (CL/F) for the parent drug only, and apparent volume of distribution (Vz/F) for the parent drug only. PK assessments were evaluated for BMS-986158 and its metabolite using plasma samples at regular intervals. Both the parent and metabolite samples were simultaneously analyzed using a validated liquid chromatography and tandem mass spectrometry assay.

Triplicate ECG measures were used to assess changes from baseline in the QT interval adjusted for heart rate using Fridericia’s formula (ΔQTcF) following a single dose of BMS-986158 at C1D1. ΔQTcF was evaluated in the context of exposure response for BMS-986158 and its metabolite of interest.

Best overall response was defined as a complete response (CR), partial response (PR), stable disease (SD), or progressive disease per RECIST 1.1 (solid tumors). ORR was defined as the number of patients who had a CR or PR divided by the total number of patients evaluated. Patients who had SD or better as their best overall response were classified as having achieved clinical benefit. These criteria must have been met at 1 time point and ≥1 subsequent time point(s) (the first of which was within 28 days of the initial time point). PFS was determined as the time from the first dose of study medication to the date of the first objective documentation of tumor progression or death due to any cause.

Summary changes from baseline in the expression of BET-regulated genes were assessed in tumor tissues and blood collected at baseline and multiple on-treatment time points after BMS-986158 administration following the 3 dosing schedules. The study also assessed summary percent changes from baseline in expression of BET-regulated genes in peripheral blood and tumors (such as *HEXIM1* and *C-MYC*). Tissue and peripheral blood samples (at baseline and on treatment) were collected for genomic, transcriptomic, and epigenetic analyses (by DNA or mRNA) to assess pharmacodynamic changes in BET-regulated genes. These samples were analyzed using whole-exome sequencing, real-time quantitative polymerase chain reaction, and RNA-sequencing methods.

### 2.4. Statistical Analyses

All treated patients who received ≥1 dose of BMS-986158 were evaluated in the safety population. To qualify for evaluation of response, patients must have had a tumor measurement at baseline and ≥1 time point after treatment, clinical progression, or death prior to their first on-treatment tumor assessment. All treated patients with available and sufficient serum or plasma samples for the corresponding analytes were included in the PK analyses. Dose proportionality of BMS-986158 and its metabolite, through regressions of log[C_max_] and/or log[AUC_0–24_] on log[dose], was assessed at single dose, and PK parameters were tabulated by combining across all schedules [21]. At multiple doses, dose proportionality was assessed separately by schedule since multiple dosing was conducted on a separate day of cycle 2 depending on schedule (e.g., day 5 at schedule A, day 14 at schedule B, and day 7 at schedule C). ECG parameters were evaluated in a categorical analysis, and linear mixed-effects regression modeling was used to describe the quantitative relationship between ΔQTcF and time-matched serum concentrations of BMS-986158 or its metabolite at a single dose.

Descriptive summary statistics, including means and 95% CIs, were calculated using the Clopper-Pearson method for ORR and Kaplan–Meier method for PFS and OS. During dose escalation, an mTPI design targeted a 27% DLT rate to guide escalation decisions and the MTD selection. In the dose-escalation segment, the target enrollment was up to 30 patients per schedule and 3–13 DLT-evaluable patients at a dose level, with an aim to determine the MTD. The totality of the data, including PK, pharmacodynamics, and patient overall safety, including at lower dose levels, was also used to select the RP2D.

All treated patients with available biomarker measurements at baseline and ≥1 time point were included in the biomarker analyses. Blood and tumor tissue samples were collected at baseline, during treatment, and at end of treatment for biomarker analysis. Markers of selected BET-regulated genes (e.g., *HEXIM1*) were analyzed with summary statistics for percent changes in expression from available matched pre- and on-treatment samples over time relative to baseline. Statistically significant changes in gene expression were determined using the Benjamini–Hochberg procedure to calculate an adjusted *p* value with an α level of 0.05.

## 3. Results

### 3.1. Patients

A total of 83 patients (schedule A, n = 46 [including 1 patient from Part 2]; schedule B, n = 8; schedule C, n = 29) were enrolled. Baseline demographic and clinical characteristics are summarized in Table 1. The median age of patients in all schedule cohorts was 59.0 years, and most enrolled patients were female (77.1%) and White (83.1%). At the time of study entry, most patients had received ≥4 prior systemic therapies (39.8%) and had a diagnosis of stage 4 cancer (85.5%). Approximately half of the patients enrolled had ovarian cancer (n = 41, 49.4%).

### 3.2. Safety

Among all treatment schedule cohorts, diarrhea was the most common TRAE at 43.4%, with all cases being grade ≤ 2 (Table 2). Thrombocytopenia occurred as a TRAE in 38.6% of all patients and was the most reported grade ≥ 3 TRAE at 25.3%. The next most common any-grade TRAEs (≥10% incidence) were fatigue (19.3%), nausea (16.9%), anemia (14.5%), decreased appetite (14.5%), and vomiting (12.0%). Treatment discontinuation due to a TRAE of grade 3 myocardial infarction occurred in 1 patient (1.2%) in the schedule A group at the 4.5-mg dose.

TRAEs had a numerically higher incidence rate in the schedule B cohort vs. the A and C cohorts (100% vs. 71.7% and 72.4%, respectively), although the patient number on schedule B (n = 8) may not have been large enough to draw a definitive conclusion about this difference. Overall, serious TRAEs occurred in 5 of 83 patients (6% of the overall study population). On schedule A, 3 of 46 patients (6.5%) experienced serious TRAEs comprising grade 4 thrombocytopenia (n = 1) as well as grade 3 myocardial infarction (n = 1) and nausea (n = 1). Serious TRAEs occurred in 2 of 8 patients (25.0%) on schedule B, including grade 3 anemia (n = 1) and grade 4 thrombocytopenia (n = 2). Among the schedule C cohort (n = 29), no serious TRAEs were reported. Among 83 patients, 9 experienced DLTs (all in part 1). The most common DLT was grade 4 thrombocytopenia, which was schedule and dose dependent (dose, n/N: schedule A—2.0 mg, 1/13; 3.0 mg, 1/10; 4.5 mg, 2/13; schedule B—3.0 mg, 2/4; schedule C—4.5 mg, 2/10). Of 4 patients, 1 experienced grade 3 nausea as a DLT in the schedule B, 2.0-mg cohort. One patient with grade 4 thrombocytopenia reported a bleeding event (grade 1 hemorrhoidal bleeding), which was considered related to the study drug.

### 3.3. BMS-986158 Pharmacokinetics

BMS-986158, as a single dose, resulted in plasma concentrations that were dose proportional from 0.75–4.5 mg and achieved peak concentrations within 1–4 h of the initial doses (Figure 1). The single dose mean T_1/2_ ranged from 34–54 h across the various doses. The metabolite of BMS-986158 achieved peak concentrations at 2–24 h and had a T_1/2_ ranging from 35–51 h across doses. Additional single-dose PK results are shown in Appendix A.

The rapid-to-moderate absorption of BMS-986158 was also observed across different doses and schedules of multidose administrations. Table 3 displays the PK parameters of BMS-986158 by multiple-dose schedule at steady state. The T_max_ across all doses and schedules ranged from 1.00–3.14 h with multiple-dose administration. In addition, over the dose range of 0.75–4.5 mg across all 3 dosing regimens, no apparent dose dependency was observed in geometric mean of CL/F at steady state, which ranged from 4.60–5.41 mL/min across doses on schedule A, 3.76–9.72 mL/min on schedule B, and 3.89–6.65 mL/min on schedule C. BMS-986158 PK profile with multiple-dose administration was approximately dose proportional on schedule A but less than dose proportional on schedule C. Dose proportionality was not assessed on schedule B because data were only available for 1 patient each at the 2.0- and 3.0-mg doses. The mean effective T_1/2_ ranged from 25.7–36.0 h (schedule A), 14.4–37.3 h (schedule B), and 27.3–31.6 h (schedule C). With multiple doses, the geometric mean AUC_(0–T)_ of BMS-986158 for schedule A ranged from 3449–29,517 h × ng/mL from 0.75–4.5-mg doses. Schedule B had geometric mean AUC_(0–T)_ values ranging from 6321–33,978 in the 2.0–3.0-mg dose groups. On schedule C, all dose groups (2.0–4.5 mg) showed less variability in AUC_(0–T)_ values (range, 18,266–19,995) due to a lower evaluated dose range. Based upon AUC_(0–24)_ at multiple doses, the PK of BMS-986158 was found to be approximately dose proportional, ranging from 2716–14,551 (schedule A), 3430–13,305 (schedule B), and 8561–11,286 (schedule C). Overall, schedule A was determined to allow continuous BMS-986158 exposure above the predicted active concentration of 46.5 ng/mL (Appendix A).

On multiple-dose schedules, the metabolite had an extended disposition (relative to BMS-986158) with an effective T_1/2_ range (depending on dose) of 72.1–80.9 h (schedule A), 35.2–105 h (schedule B), and 63.5–72.8 h (schedule C; Table 3). The geometric mean metabolite-to-parent AUC ratio (MR AUC_[0_–_T]_) ranged from 0.14–0.23 h × ng/mL (schedule A), 0.16–0.22 h × ng/mL (schedule B), and 0.19–0.29 h × ng/mL (schedule C). In addition, the metabolite appeared to be approximately dose proportional, as the MR AUC_(0–24)_ showed a narrow range in each treatment schedule (schedule A, 0.13–0.20; schedule B, 0.15–0.19; schedule C, 0.15–0.23).

When cardiac parameters were evaluated at single dose, neither BMS-986158 nor its metabolite had a statistically significant effect on ΔQTcF. The highest observed C_max_ of BMS-986158 or its metabolite was predicted to have a small increase in ΔQTcF (<5 msec), with 2-sided 90% CI upper bounds of <7 msec, which is below the threshold of concern (1-sided 95% CI upper bound of 10.0 msec).

### 3.4. Efficacy

Preliminary efficacy data are presented in Table 4 and Appendix A. Overall, 31.3% (n = 26) of patients experienced SD or better across treatment schedules and doses. No patients experienced a CR, and 2 patients on schedule A had a PR (1 patient on the 4.5-mg dose, and 1 patient on the 4.5-mg dose for 2 weeks followed by 3.75 mg for the remainder of the study), for ORRs of 2% on schedule A and 0% on schedules B and C. SD was achieved in 26.1% (schedule A), 37.5% (schedule B), and 31.0% (schedule C) of patients. Approximately half (50.0–52.2% across the treatment schedules) of the patients had progressive disease as best response. The best overall response was unable to be determined in 17.4% (schedule A), 12.5% (schedule B), and 17.2% (schedule C) of patients.

Appendix A display the baseline characteristics and treatment assignments of patients who achieved clinical benefit (i.e., SD or better as best overall response). Although the patient numbers were insufficient for a subgroup analysis, some qualitative differences were observed. Of the patients who received the 4.5-mg dose on schedule A, 64% (9/14) experienced clinical benefit. Tumor types in which over half the patients achieved clinical benefit included NUT carcinoma (57.1%, 4/7), uveal melanoma (60%, 3/5), and adenoid cystic carcinoma (100%, 4/4). The 2 patients with a PR on schedule A had ovarian cancer (4.5-mg dose) and NUT carcinoma (4.5-mg/3.75-mg dose). Figure 2 shows radiographs from a 66-year-old woman with stage IIIC serous papillary ovarian cancer harboring somatic *BRCA2* and *TP53* mutations. Following initial diagnosis and tumor resection (prior to study entry), the patient was treated with 6 cycles of carboplatin-paclitaxel as a first-line therapy. She experienced recurrence 17 months later with lesions in the retroperitoneal and mediastinal lymph nodes, which were rechallenged with 6 cycles of carboplatin-paclitaxel and another 6 cycles, 24 months later, which resulted in a PR. Over the next approximately 8 years, tumor recurrence was treated with doxorubicin, topotecan, carboplatin, gemcitabine, or an investigational bispecific anti–mesothelin-CD40 antibody before progressive disease occurred prior to enrollment in the current trial reported here. In the current study, the patient was assigned to receive BMS-986158 4.5 mg on schedule A. Prior to the C1D1 visit, the patient had elevated levels of the tumor marker cancer antigen (CA) 125 (3032 U/mL), which decreased and remained stable after treatment initiation (Figure 2B). The patient achieved a PR at day 170, with a 35% reduction in tumor burden. She had a PR for 155 days on study treatment until she experienced clinical progression.

Overall, the median (95% CI) PFS was 8.29 (7.14–9.71) weeks on schedule A, 8.43 (5.57–40.14) weeks on schedule B, and 9.00 (7.86–13.29) weeks on schedule C. The PFS rates (95% CI) at 24 and 48 weeks, respectively, were 18.1% (8.0–31.5%) and 10.3% (3.3–22.1%) on schedule A, 42.9% (9.8–73.4%) and 14.3% (0.7–46.5%) on schedule B, and 13.5% (3.4–30.6%) and 6.8% (0.6–24.4%) on schedule C. The OS rates at 6 and 12 months, respectively, were 52.6% (37.0–66.0%) and 34.3% (20.8–48.2%) on schedule A and 48.4% (27.3–66.7%) and 30.8% (13.7–49.7%) on schedule C. OS rates were not reported on schedule B due to insufficient patient numbers.

### 3.5. Biomarker Analyses

Changes in gene expression were evaluated in 41 patients, using peripheral whole blood samples (collected at baseline and 8 h after BMS-986158 treatment) on C1D1. Based on a nonlinear model of >18,000 genes measured for transcriptional changes in matched pre- and post-treatment biopsies, 4500 showed significant changes in expression in patients treated with BMS-986158 (Benjamini–Hochberg–adjusted *p* < 0.05). *CCR2* showed transcriptional suppression during BMS-986158 treatment and was maintained on days off treatment (Figure 3). *CCR2* mRNA had dose-dependent expression changes to single-dose BMS-986158 up to 4.5 mg. *HEXIM1* mRNA, an established BET inhibitor pharmacodynamic biomarker [22], was dose dependently upregulated by BMS-986158.

## 4. Discussion

Over 15 BET inhibitors have been in early phase clinical development across a variety of cancer types [23], and, as a whole, have shown modest efficacy in clinical trials [24,25].

BMS-986158 is a highly potent and selective BET inhibitor that has shown preclinical antitumor activity in hematologic and solid tumors [19]. Our phase 1/2a trial of BMS-986158 evaluated safety and preliminary clinical efficacy in patients with advanced solid tumors. On schedule A, the most common TRAEs (defined as occurring in ≥10% of patients) were thrombocytopenia, diarrhea, fatigue, nausea, anemia, decreased appetite, and vomiting, and serious TRAEs occurred in 6.0% of all patients. These TRAEs are typical within the BET inhibitor class; a recent meta-analysis of BET inhibitors found that the most common AEs included thrombocytopenia, anemia, neutropenia, gastrointestinal symptoms, and fatigue [24]. The only grade 3/4 TRAE reported in our trial with an incidence of ≥5% on schedule A treatment was thrombocytopenia.

Schedule A provided more continuous drug exposure than schedules B and C, and a C_max_ value comparable to that on schedule B. The results from Part 1 of this study supported the more continuous dosing of schedule A in the dose expansion. Study CA011-001 was terminated early after 1 patient enrolled in Part 2 of the study and received 4.5 mg for 2 weeks followed by 3.75 mg for the remainder of the trial. The study termination was due to a business decision and not related to safety reasons. As no additional patients were enrolled in Part 2 prior to study termination, the aforementioned patient is included in the schedule A, 4.5-mg dose cohort in this report. On schedule A, BMS-986158 exhibited a dose-proportional PK profile. Rapid absorption and slow elimination were observed on all schedules. In addition, *CCR2* transcriptional suppression and *HEXIM1* transcriptional activation were observed after BMS-986158 dosing, showing an on-target pharmacodynamic response and inhibition of the BET pathway, since *HEXIM1* [26,27] and *CCR2* mRNA [28] have been shown to be upregulated and downregulated, respectively, in tumor cell lines treated with BET inhibitors.

We also assessed the PK of a disproportionate metabolite of interest for BMS-986158, which demonstrated approximately dose-proportional kinetics on schedule A. The metabolite showed generally consistent ranges in MR C_max_ (11–17%) and AUC_(0–T)_ (14–29%) at multiple doses across all the clinically evaluated dosing schedules. This demonstrates that the tested metabolite may be a major metabolite of interest. The metabolite achieved lower relative plasma concentrations than the parent drug, BMS-986158, while having a relatively longer effective T_1/2_ at multiple doses, as would be expected for a metabolite exhibiting elimination rate–limited metabolite kinetics. In addition, single doses of BMS-986158 were not found to cause significant QTc prolongation within the tested dose range.

Although this study was primarily focused on safety, some indications of clinical activity were observed. On schedule A, 30.4% of patients achieved clinical benefit (a well-accepted measure of disease stabilization in oncology clinical trials [29]) across doses and treatment schedules, and 2 patients had a PR (1 with ovarian cancer treated with the 4.5-mg dose and 1 with NUT carcinoma treated with 4.5 mg for 2 weeks followed by 3.75 mg). This rate of clinical benefit was within the range of what was reported in a systematic review of 12 BET inhibitors, which showed an overall SD rate of 27.4% (range, 8.0–60.0%) and a PR rate of 5.7% (range, 2.0–21.0%) [24].

Trends observed in our study suggested that a higher proportion of patients experienced clinical benefit among those who received 4.5 mg on schedule A (compared with all tested doses on schedule B or C) or had baseline NUT carcinoma, uveal melanoma, or adenoid cystic carcinoma (relative to other cancer types). As demonstrated in a patient case, a female patient with ovarian cancer receiving BMS-986158 4.5 mg on schedule A experienced a reduction of 35% in tumor burden by day 170 after treatment initiation, providing some preliminary evidence of potential efficacy of BMS-986158. In addition, the trend that patients with baseline NUT carcinoma experienced clinical benefit is consistent with antitumor activity found among patients with NUT carcinoma treated with the BET inhibitors molibresib, birabresib, and OTX015 [18,23], which may be indicative of altered BET inhibitor interactions with BRD3-NUT, BRD4-NUT, or NSD3-NUT fusions. Preclinical research has demonstrated inhibition of gene expression and antitumor effects with BET inhibitors in uveal melanoma [14] and adenoid cystic carcinoma cell lines [30], with preliminary dose-dependent efficacy found in uveal melanoma in humans [31] The antitumor effect of a BET inhibitor in uveal melanoma [14] or adenoid cystic carcinoma [30] has been postulated to occur via cell cycle arrest due to inhibition of BRD4.

The PFS ranged from 8.29–9.00 weeks depending on treatment schedule. The PFS rates at 24 and 48 weeks ranged from 13.5–42.9% and 6.8–14.3%, respectively, depending on treatment schedule. Due to the population heterogeneity and other limitations of a dose-escalation trial, future studies will be required to confirm any potential antitumor activity in patients with these histologies and treatment schedule.

## 5. Conclusions

BMS-986158 had a well-tolerated safety profile on schedule A (5 days on, 2 days off), with the most common TRAE being thrombocytopenia. PK was dose proportional, favoring a more continuous dosing approach (schedule A) over intermittent schedules (B and C). Preliminary efficacy results indicated antitumor activity noted by 30.4% of patients on schedule A achieving clinical benefit. Currently BMS-986158 is under investigation in a phase 1b/2 study evaluating combination therapy of BMS-986158 with the Janus kinase inhibitors ruxolitinib or fedratinib in patients with myelofibrosis (NCT04817007) [32].

## Figures and Tables

**Figure 1 cancers-14-04079-f001:**
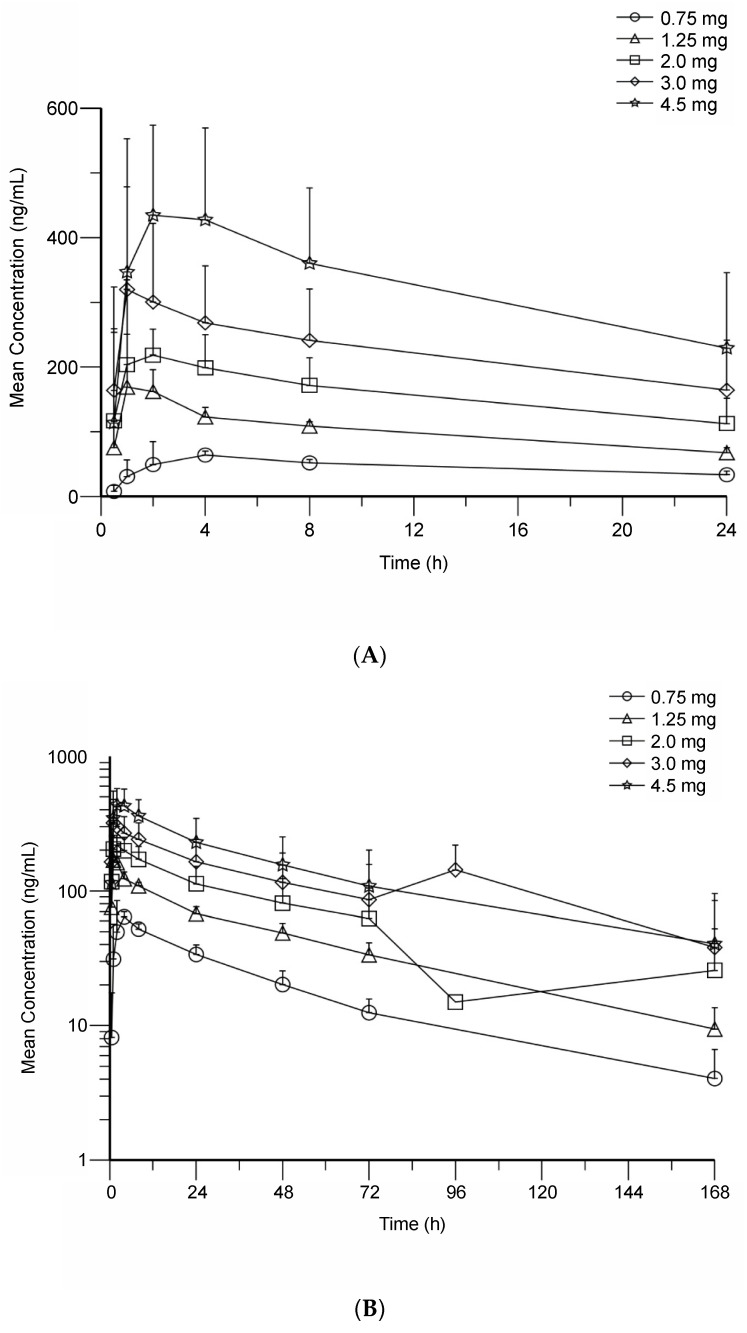
Mean plasma concentration, following a single dose of BMS-986158, of (**A**) BMS-986158 through 24 h; (**B**) BMS-986158 through 168 h; (**C**) a metabolite of BMS-986158 through 24 h; and (**D**) the metabolite through 168 h.

**Figure 2 cancers-14-04079-f002:**
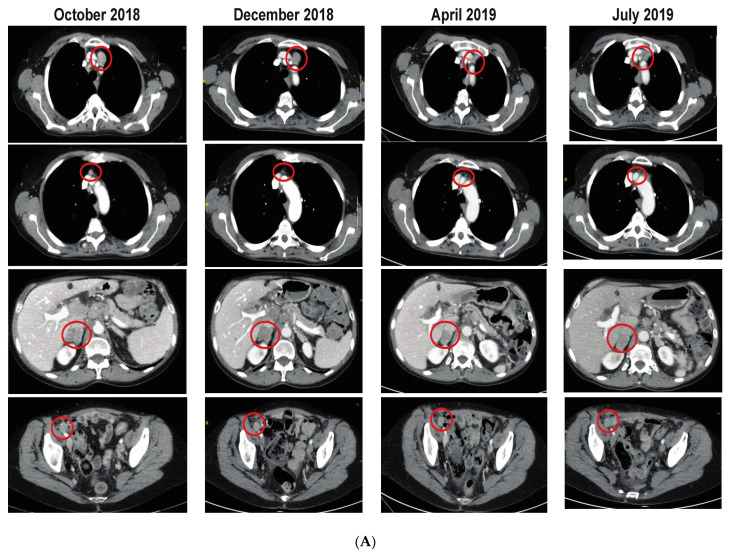
BMS-986158 4.5-mg treatment on schedule A in a 66-year-old female patient with serous papillary ovarian cancer. (**A**) The patient had tumors in the lymph nodes (top 2 rows), adrenal glands (third row), and peritoneum (bottom row), with tumor shrinkage observed in the lymph nodes. (**B**) The patient had elevated levels of the tumor marker cancer antigen (CA) 125 at C1D1 (5 November 2018), which decreased and stabilized. Dates are shown as month-day-year.

**Figure 3 cancers-14-04079-f003:**
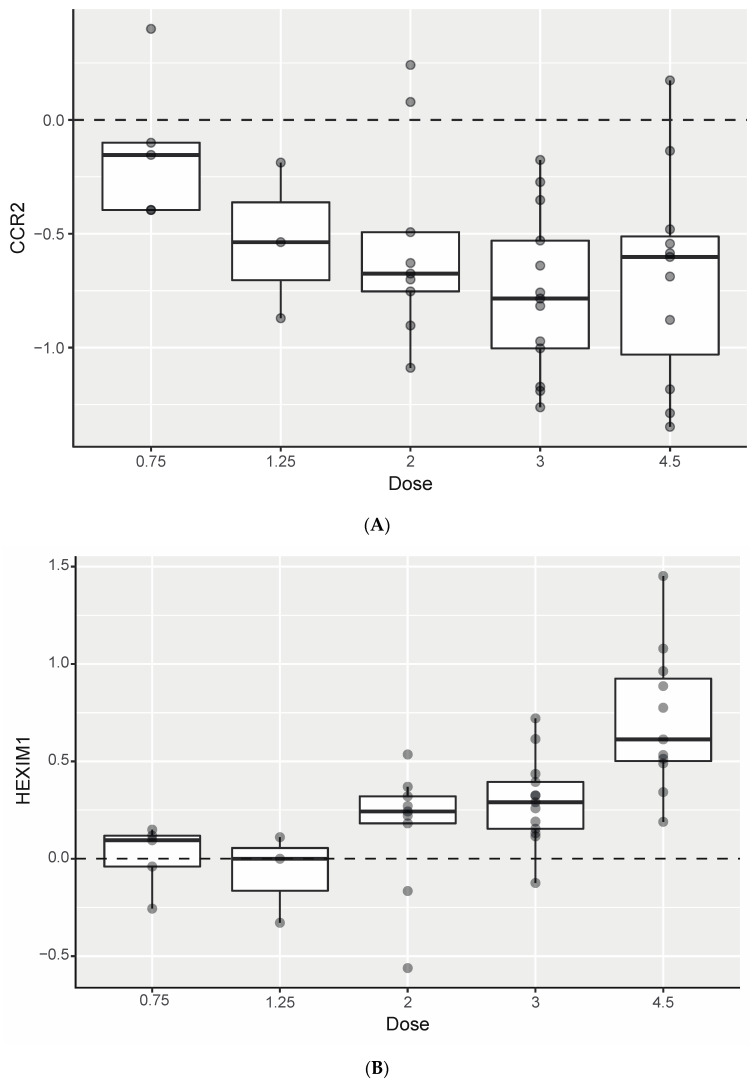
Changes from baseline in gene expression in tumor biopsies by dose on schedule A in (**A**) *CCR2* and (**B**) *HEXIM1*. Center lines in the boxes represent the median value, the top and bottom of the boxes show the interquartile range (IQR), the vertical bars indicate 1.5 × IQR, and each dot represents one sample.

**Table 1 cancers-14-04079-t001:** Baseline clinical and demographic characteristics by treatment schedule for BMS-986158 monotherapy QD ^a^.

	Demographic Characteristics and Prior Treatments			
	Schedule A (5 Days On, 2 Days Off) n = 46	Schedule B (14 Days On, 7 Days Off) n = 8	Schedule C (7 Days On, 14 Days Off) n = 29	Total N = 83
**Age, median (range), years**	57.0 (23–88)	64.0 (58–75)	57.0 (33–71)	59.0 (23–88)
**Sex, n (%)**				
**Male**	10 (21.7)	3 (37.5)	6 (20.7)	19 (22.9)
**Female**	36 (78.3)	5 (62.5)	23 (79.3)	64 (77.1)
**Race, n (%)**				
**White**	39 (84.8)	8 (100.0)	22 (75.9)	69 (83.1)
**Black**	2 (4.3)	0	0	2 (2.4)
**Asian**	2 (4.3)	0	2 (6.9)	4 (4.8)
**Other**	3 (6.5)	0	5 (17.2)	8 (9.6)
**Prior systemic therapies, n (%)**				
**0**	2 (4.3)	2 (25.0)	1 (3.4)	5 (6.0)
**1**	17 (37.0)	0 (0)	5 (17.2)	22 (26.5)
**2**	3 (6.5)	2 (25.0)	4 (13.8)	9 (10.8)
**3**	7 (15.2)	2 (25.0)	5 (17.2)	14 (16.9)
**≥4**	17 (37.0)	2 (25.0)	14 (48.3)	33 (39.8)
	**Disease characteristics**			
**Current disease stage, n (%)**				
**1**	0	0	1 (3.4)	1 (1.2)
**2**	1 (2.2)	0	0	1 (1.2)
**3**	5 (10.9)	1 (12.5)	4 (13.8)	10 (12.0)
**4**	40 (87.0)	7 (87.5)	21 (82.8)	71 (85.5)
**Tumor type, n (%)**				
**Ovarian cancer**	20 (43.5)	3 (37.5)	18 (62.1)	41 (49.4)
**SCLC**	5 (10.9)	2 (25.0)	3 (10.3)	10 (12.0)
**TNBC**	6 (13.0)	1 (12.5)	2 (6.9)	9 (10.8)
**NUT carcinoma**	6 (13.0)	0	1 (3.4)	7 (8.4)
**UM**	4 (8.7)	0	1 (3.4)	5 (6.0)
**ACC**	1 (2.2)	1 (12.5)	2 (6.9)	4 (4.8)
**CRPC**	1 (2.2)	0	1 (3.4)	2 (2.4)
**Ewing sarcoma**	1 (2.2)	0	0	1 (1.2)
**Metastatic high-grade NEC**	0	0	1 (3.4)	1 (1.2)
**PPSC**	1 (2.2)	0	0	1 (1.2)
**SpCC**	0	1 (12.5)	0	1 (1.2)
**UCS**	1 (2.2)	0	0	1 (1.2)

^a^ All study sites participated in all dosing schedules. ACC, adenoid cystic carcinoma; CRPC, castration-resistant prostate cancer; NUT, nuclear protein in testis; NEC, neuroendocrine carcinoma; PPSC, primary peritoneal serous carcinoma; QD, once daily; SCLC, small cell lung cancer; SpCC, spindle cell carcinoma; TNBC, triple-negative breast cancer; UCS, uterine carcinosarcoma; UM, uveal melanoma.

**Table 2 cancers-14-04079-t002:** Safety summary of TRAEs by dosing schedule.

	BMS-986158 Monotherapy QD							
	Schedule A (5 Days On, 2 Days Off) n = 46		Schedule B (14 Days On, 7 Days Off) n = 8		Schedule C (7 Days On, 14 Days Off) n = 29		Total N = 83	
	Any Grade n (%)	Grade ≥ 3 n (%)	Any Grade n (%)	Grade ≥ 3 n (%)	Any Grade n (%)	Grade ≥ 3 n (%)	Any Grade n (%)	Grade ≥ 3 n (%)
**Patients with any TRAE**	33 (71.7)	19 (41.3)	8 (100.0)	4 (50.0)	21 (72.4)	5 (17.2)	62 (74.7)	28 (33.7)
**TRAEs in ≥10% of all pts**								
**Diarrhea**	19 (41.3)	0	4 (50.0)	0	13 (44.8)	0	36 (43.4)	0
**Thrombocytopenia**	20 (43.5)	16 (34.8)	5 (62.5)	2 (25.0)	7 (24.1)	3 (10.3)	32 (38.6)	21 (25.3)
**Fatigue**	8 (17.4)	2 (4.3)	4 (50)	1 (12.5)	4 (13.8)	0	16 (19.3)	3 (3.6)
**Nausea**	7 (15.2)	1 (2.2)	1 (12.5)	1 (12.5)	6 (20.7)	0	14 (16.9)	2 (2.4)
**Anemia**	6 (13.0)	2 (4.3)	2 (25.0)	2 (25.0)	4 (13.8)	1 (3.4)	12 (14.5)	5 (6.0)
**Decreased appetite**	5 (10.9)	0	4 (50.0)	0	3 (10.3)	0	12 (14.5)	0
**Vomiting**	5 (10.9)	0	2 (25.0)	1 (12.5)	3 (10.3)	1 (3.4)	10 (12.0)	2 (2.4)
**Serious TRAEs**	3 (6.5) ^a,d^	3 (6.5) ^a,d^	2 (25.0) ^b^	2 (25.0) ^b^	0	0	5 (6.0) ^c,d^	5 (6.0) ^c,d^
**Discontinuations due to TRAEs**	1 (2.2) ^d^	1 (2.2) ^d^	0	0	0	0	1 (1.2) ^d^	1 (1.2) ^d^

^a^ Serious TRAEs were grade 4 thrombocytopenia (n = 1), and grade 3 myocardial infarction (n = 1) and nausea (n = 1); ^b^ Serious TRAEs were grade 4 thrombocytopenia (n = 2) and grade anemia (n = 1); ^c^ Serious TRAEs were grade 4 thrombocytopenia (n = 3), grade 3 anemia (n = 1), grade 3 myocardial infarction (n = 1), and grade 3 nausea (n = 1); ^d^ Reason for discontinuation was grade 3 myocardial infarction (n = 1). pt, patient; QD, once daily; TRAE, treatment-related adverse event.

**Table 3 cancers-14-04079-t003:** Pharmacokinetics of BMS-986158 and its metabolite at steady state after multiple-dose treatment schedule.

Pharmacokinetic Parameter	Schedule A (5 Days On, 2 Days Off)					Schedule B (14 Days On, 7 Days Off)		Schedule C (7 Days On, 14 Days Off)		
	0.75 mg n = 4	1.25 mg n = 4	2.0 mg n = 7	3.0 mg n = 8	4.5 mg n = 8	2.0 mg n = 1	3.0 mg n = 1	2.0 mg n = 5	3.0 mg n = 8	4.5 mg n = 7
	**BMS-986158**									
**C_max_ (ng/mL), GM (%CV)**	136 (43)	284 (16)	442 (29)	624 (44)	898 (39)	279	855	520 (34)	588 (47)	901 (62)
**T_max_ (h), median** **(range)**	3.14 (2.00–6.05)	1.50 (0.50–2.00)	2.00 (0.50–4.02)	2.01 (0.50–2.10)	2.00 (1.00–4.05)	1.00	1.00	1.00 (0.83–4.00)	1.66 (0.97–2.03)	2.00 (0.17–2.03)
**AUC_(0–24)_ (h × ng/mL), GM (%CV)**	2716 (51) ^a^	3852 ^b^	N/A ^c^	9817 (56) ^d^	14,551 (32)	3430	13,305	8561 (36)	8637 (71)	11,286 (104)
**AUC_(0–T)_ (h × ng/mL), GM (%CV)**	3449 (93)	4961 (73)	7612 (71)	13,378 (77)	29,517 (36)	6321	33,978	19,868 (45)	18,266 (97)	19,995 (135)
**Effective T_1/2_ (h), mean (SD)**	36.0 (16.0) ^a^	N/A ^c^	N/A ^c^	25.7 (14.9) ^d^	27.3 (8.7)	14.4	37.3	31.6 (10.5)	27.3 (15.2) ^d^	27.6 (19.1) ^e^
**CL/F (mL/min), GM (%CV)**	4.60 (55) ^a^	5.41 ^b^	N/A^c^	5.09 (43) ^d^	5.15 (29)	9.72	3.76	3.89 (57)	5.79 (39)	6.65 (60)
	**Metabolite of BMS-986158**									
**C_max_ (ng/mL), GM (%CV)**	25.8 (82) ^a^	31.0 (25)	49.4 (46)	83.3 (57)	126 (50)	32.7	127	80.7 (53)	64.5 (55)	146 (83)
**MR C_max_, GM (%CV)**	0.17 (41) ^a^	0.11 (29)	0.11 (25)	0.13 (37)	0.14 (18)	0.12	0.15	0.16 (33)	0.11 (32)	0.16 (33)
**T_max_ (h), median (range)**	24.0 (24.0–24.0)^a^	4.00 (1.50–6.00)	2.07 (0–4.02)	2.01 (1.00–6.10)	4.00 (1.00–24.0)	1.00	1.00	4.08 (3.83–24.0)	4.00 (0–6.32)	4.00 (1.00–27.1)
**MR AUC_(0–24)_ (h × ng/mL), GM (%CV)**	0.20 (37) ^a^	0.13 ^b^	N/A^c^	0.18 (36) ^d^	0.18 (25)	0.15	0.19	0.21 (35)	0.15 (35)	0.23 (45)
**MR AUC_(0–T)_ (h × ng/mL), GM (%CV)**	0.23 (40) ^a^	0.14 (18)	0.15 (25)	0.22 (34)	0.23 (24)	0.16	0.22	0.25 (43)	0.19 (39)	0.29 (48)
**Effective T_1/2_ (h), mean (SD)**	80.9 (40.4) ^a^	N/A ^c^	N/A ^c^	72.1 (47.1) ^d^	80.4 (29.6)	35.2	105	72.7 (19.4)	63.5 (39.5)	72.8 (74.2)

^a^ n = 3; ^b^ n = 1; ^c^ n = 0, ^d^ n = 7; ^e^ n = 5. AUC_(0–24)_, area under the plasma concentration-time curve from time 0 to time 24 h post dose; AUC_(0–T)_, area under the plasma concentration-time curve from time 0 to time of the last quantifiable concentration; CL/F, apparent total body clearance; C_max_, maximum observed plasma concentration; %CV, coefficient of variation; GM, geometric mean; MR, metabolite-to-parent; N/A, not available; T_1/2_, half-life; T_max_, time to C_max_.

**Table 4 cancers-14-04079-t004:** Preliminary efficacy by dosing schedule.

	Schedule A n = 46	Schedule B n = 8	Schedule C n = 29
**Best overall response by investigator, n (%)**			
**Complete response**	0	0	0
**Partial response**	2 (4.3) ^a^	0	0
**Stable disease**	12 (26.1)	3 (37.5)	9 (31.0)
**Progressive disease**	24 (52.2)	4 (50.0)	15 (51.7)
**Unable to determine**	8 (17.4) ^b^	1 (12.5) ^c^	5 (17.2) ^d^
**ORR (95% CI), %**	2 (0.5–14.8)	0 (0.0–36.9)	0 (0.0–11.9)
**Median PFS (95% CI), weeks**	8.29 (7.14–9.71)	8.43 (5.57–40.14)	9.00 (7.86–13.29)
**PFS rate, n (95% CI)**			
**24 weeks**	18.1 (8.0–31.5)	42.9 (9.8–73.4)	13.5 (3.4–30.6)
**48 weeks**	10.3 (3.3–22.1)	14.3 (0.7–46.5)	6.8 (0.6–24.4)
**Median OS (95% CI), months**	7.52 (4.17–11.99)	9.23 (2.79–11.83)	5.32 (3.52–11.83)
**OS rate, n (95% CI)**			
**6 months**	52.6 (37.0–66.0)	NR ^e^	48.4 (27.3–66.7)
**12 months**	34.3 (20.8–48.2)	NR ^e^	30.8 (13.7–49.7)

^a^ Partial responses observed in 1 patient with ovarian cancer treated with 4.5 mg BMS-986158 and 1 patient with NUT carcinoma treated with 4.5 mg/3.75 mg; ^b^ Reasons unable to determine: death prior to disease assessment (n = 4) and other (n = 4); ^c^ Reason unable to determine: other (n = 1); ^d^ Reasons unable to determine: death prior to disease assessment (n = 1) and other (n = 4); ^e^ OS rate not reported for schedule B due to insufficient patient number. NR, not reported; NUT, nuclear protein in testis; ORR, objective response rate; OS, overall survival; PFS, progression-free survival.

## Data Availability

Data may be shared upon request to the corresponding author.

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
