# Peer review of "BMS-986158, a Small Molecule Inhibitor of the Bromodomain and Extraterminal Domain Proteins, in Patients with Selected Advanced Solid Tumors: Results from a Phase 1/2a Trial"

_cancers, 2022, doi:10.3390/cancers14174079_

Round 1

Reviewer 1 Report

This is a P1/2a trial of a novel BET inhibitor, BMS-986158, in patients with advanced cancer. There were 2 parts, Part one was a dose finding analysis and part 2 was the expansion cohort at the RP2D. This was a well written manuscript and the data is presented in an organized and concise manner. 

My comments and concerns are below:

1)    Part 2 of the study, the expansion cohort:

a.     In the Methods section, 2.1 Study Design, there is no description of Part 2 of the study. It should be included. It is not clear to me whether the fusion proteins or amplifications were required for this part of the study.

b.     Also, it seems that there was only one patient enrolled to Part 2. Why is this? I take it was because the study was terminated prematurely. This should be explained further in the results and the discussion.

2)    In the Methods section they stated that Part 1 only included patients with ovarian cancer, SCLC, or TNBC.  If only one patient was enrolled to Part 2, why are there so many other cancer types, CRPC, Ewings, NEC, etc. This is also depicted in Figure S1.  There needs to be some clarification of this.

3)    In the methods it stated, “Dose escalation was guided by a modified toxicity probability interval (mTPI) design.” This needs a reference.

4)    The authors’ often cite the clinical benefit rate, although this is not stated as one of their endpoints. It may be more prudent to focus on their defined endpoint of ORR.  Given the modest ORR observed, the authors should comment in the discussion if they would move forward with this drug as  a  single agent and/or propose other treatment combinations/cancer settings to enhance response.

5)    There is a problem with figure S4.  The fraction for the Schedule B 3.0mg dose is missing (I suspect ¼).

Reviewer 2 Report

Hilton and coworkers reported results from a phase 1/2a trial with BMS-986158, a BET inhibitor. The 7-year clinical trial NCT02419417 with actual enrollment 83 participants provided the safety, tolerability, PK and pharmacodynamics of BMS-986158. Overall this study was well designed and has been carried out with care and the SI looks professional. So the research is suggested to be published in cancers in present form.
